# The Role of Hypofractionation in Proton Therapy

**DOI:** 10.3390/cancers14092271

**Published:** 2022-05-02

**Authors:** Alexandre Santos, Scott Penfold, Peter Gorayski, Hien Le

**Affiliations:** 1Australian Bragg Centre for Proton Therapy and Research, North Terrace, Adelaide, SA 5000, Australia; scott.penfold@sahmri.com (S.P.); peter.gorayski@sa.gov.au (P.G.); hien.le@sahmri.com (H.L.); 2Department of Radiation Oncology, Royal Adelaide Hospital, Port Road, Adelaide, SA 5000, Australia; 3School of Physical Sciences, The University of Adelaide, North Terrace, Adelaide, SA 5005, Australia; 4Division of Health Sciences, University of South Australia, North Terrace, Adelaide, SA 5000, Australia

**Keywords:** hypofractionation, proton therapy, high dose, high dose per fraction

## Abstract

**Simple Summary:**

Hypofractionation is a radiation oncology concept in which larger doses per fraction are administered and therefore less fractions are prescribed in a given treatment course. This concept has been increasingly utilized in recent years in external beam radiotherapy (EBRT). Hypofractionation has the potential benefits of reducing the length of treatment and also treatment costs. In the case of proton beam therapy (PBT), which is a limited resource with only 99 facilities currently in operation worldwide in 2021, hypofractionation also has the benefit of potentially increasing the number of patients able to access PBT. This review article will discuss the reported clinical data and radiobiology of hypofractionated PBT. The aim is to report on the current knowledge of hypofractionation in PBT and the issues which still require investigation. Over 50 published manuscripts reporting clinical results involving hypofractionation and PBT were included in this review, where the most common treatment regions were prostate, lung and liver. Future clinical trials are still needed to determine the optimal fractionation regime.

**Abstract:**

Hypofractionated radiotherapy is an attractive approach for minimizing patient burden and treatment cost. Technological advancements in external beam radiotherapy (EBRT) delivery and image guidance have resulted in improved targeting and conformality of the absorbed dose to the disease and a reduction in dose to healthy tissue. These advances in EBRT have led to an increasing adoption and interest in hypofractionation. Furthermore, for many treatment sites, proton beam therapy (PBT) provides an improved absorbed dose distribution compared to X-ray (photon) EBRT. In the past 10 years there has been a notable increase in reported clinical data involving hypofractionation with PBT, reflecting the interest in this treatment approach. This review will discuss the reported clinical data and radiobiology of hypofractionated PBT. Over 50 published manuscripts reporting clinical results involving hypofractionation and PBT were included in this review, ~90% of which were published since 2010. The most common treatment regions reported were prostate, lung and liver, making over 70% of the reported results. Many of the reported clinical data indicate that hypofractionated PBT can be well tolerated, however future clinical trials are still needed to determine the optimal fractionation regime.

## 1. Introduction

Proton beam therapy (PBT) is an advanced form of external beam radiotherapy (EBRT) with the potential to increase the therapeutic ratio relative to conventional radiotherapy with X-rays (photons) for a range of indications. However, the large capital expenditure required to establish a proton therapy facility has and will continue to limit the availability of this technology. Reducing the cost to deliver PBT is an active area of research that includes technological concepts such as more compact particle accelerators and gantry-free treatment solutions [1]. An alternative strategy for improving access to PBT is to reduce the number of treatment fractions per course, thereby increasing the number of patients that can be treated at a given facility.

Furthermore, as PBT is a limited resource, many patients are required to travel to access the technology. Reducing the length of time a patient is away from their home and their extended support network can have significant financial and psychosocial implications [2]. Therefore, hypofractionation of PBT is an important area of research in several respects.

Australia’s first proton therapy centre is currently under construction in Adelaide. Due to the size of Australia and the geographic distribution of its population, the majority of patients being treated at the centre will be required to travel a large distance for their treatment course. Also, being the only PBT centre in the country for at least the short-term, maximizing access to this technology is critical. Therefore, hypofractionation is an area of particular interest in the Australian context.

The purpose of this review is to summarize the current clinical evidence related to hypofractionation in PBT. The last similar review on this topic was performed by Laine et al. in 2016 [3], however that study included photons and heavier ions, and was limited to extracranial treatment sites. The current review will focus specifically on PBT, include all treatment sites and provide an update on studies published since the review by Laine et al.

## 2. Radiobiology

When the α/β ratio is high for the tumor tissue and low for surrounding late normal tissue complications, delivering radiotherapy in many small doses per fraction should yield the highest therapeutic ratio. However, as the α/β ratio for the tumor is reduced relative to the surrounding healthy tissue, the benefit of a larger number of fractions similarly reduces [4].

PBT has the advantage of reducing the dose to healthy tissue compared to X-ray techniques due to the Bragg peak. Additionally, protons have radiobiological differences compared with X-rays. Near the end of the proton range the density of ionization events, and subsequently linear energy transfer (LET), increases. This increase in LET results in more complex DNA damage, which correspondingly reduces a cell’s ability to repair that damage. It is common clinical practice to utilize a constant relative biological effectiveness (RBE) of 1.1 for protons. That is, a given proton physical dose is assumed to be 10% more radiobiologically damaging than the same photon physical dose. While this assumption has been widespread in the clinic and is recommended to be maintained by the American Association of Physicists in Medicine (AAPM, Alexandria, VA, USA), our knowledge of RBE does not support a constant value independent of α/β ratio, fraction size or LET [5].

In the case of hypofractionation, as the dose per fraction increases, one would expect a decrease in the RBE of protons [6]. This decrease in RBE is due to the shape of the cell survival curves for protons and photons. At lower single doses, the cell survival curve is typically steeper for proton radiation than photon radiation, while at larger single doses, the gradients of the cell survival curves are more equivalent, resulting in a smaller relative separation between protons and photons.

A recent review of published experimental data by Paganetti included 76 publications with 369 data points to estimate the RBE for cell survival [7]. It was found that the average RBE across all α/β does increase with LET from ~1.1 in the entrance region, to ~1.15 in the center, ~1.35 at the distal edge and ~1.7 in the distal fall-off in a spread out Bragg peak. These values were for 2 Gy irradiations, and the RBE decreased by up to 10% for 6 Gy irradiations.

## 3. Clinical Results of Hypofractionation with Protons

A Pubmed search of keywords (“proton” AND “hypofractionation”) OR (“proton” AND “hypofractionated”) OR (“proton” AND “hypofraction”) was performed on the 17 July 2021. For the purposes of this study, hypofractionation was considered as a treatment delivery regimen of greater than 2 Gy (RBE) per fraction. The list of publications was reviewed and those not reporting clinical data were omitted. Some manuscripts were later identified which reported clinical outcomes of dose fractionations greater than 2 Gy (RBE) per fraction and were included in the review. The search and subsequent review of eligibility resulted in the inclusion of 57 publications. Figure 1 shows the percentage of the included publications as a function of publication year. Since the first study published in 2001, there has been a significant increase in the number of studies over time, with approximately 90% published since 2010.

Figure 2 shows a breakdown of included publications by treatment region. The most common were lung, prostate and liver, which made up 74% of the published works. The other reported treatment regions were brain, breast, head and neck (H&N), central nervous system (CNS) and sarcomas.

### 3.1. Prostate

Over the past 10 years there have been a significant number of clinical trials and studies reporting results on hypofractionated PBT for prostate cancer. A range of doses per fraction have been investigated, ranging from 2.5 to 7.6 Gy (RBE). Table 1 summarizes the reported clinical results.

In 2012 Johansson et al. reported on the five-year outcome of the use of a hypofractionated proton boost to be an alternative to high dose rate (HDR) brachytherapy [8]. 265 patients were prescribed 22 Gy (RBE) in four fractions using a single perineal proton beam, followed after one-week rest by X-ray EBRT of 50 Gy in daily 2 Gy fractions. In 2019, Johansson et al. reported on an update to their results, now with 531 patients treated between 2002 to 2015 [9]. The patients were considered to be low-risk (18.7%), intermediate-risk (31.3%), high-risk (26.8%) and very high-risk (23.2%). The five and 10-year biochemical control rates were 100% and 94% in low-risk patients; 94% and 87% in intermediate-risk patients; 82% and 63% in high-risk patients; and 72% and 55% in very high-risk patients, respectively.

In 2013 Kim et al. reported on a phase II randomized trial which assigned 82 patients to one of five different fractionation schedules (ClinicalTrials.gov, Bethesda, MD, USA, no. NCT01709253) [10]. These were either Arm 1, 60 Gy (RBE) in 20 fractions over five weeks; Arm 2, 54 Gy (RBE) in 15 fractions over five weeks; Arm 3, 47 Gy (RBE) in 10 fractions over five weeks; Arm 4, 35 Gy (RBE) in five fractions over 2.5 weeks; or Arm 5, 35 Gy (RBE) in five fractions over five weeks. Arms 1–3 were referred to as moderately hypofractionated (MHF) and Arms 4 and 5 as extreme hypofractionated (EHF). The patients consisted of low-risk (34%), intermediate-risk (45%), and high-risk (21%). More recent longer-term results of the study were reported by Ha et al. in 2019 [11]. The seven-year biochemical control rates of the MHF and EHF groups were 90.5 and 57.1% in the low-risk group; 83.5 and 42.9% in the intermediate-risk group; and 41.7 and 40.0% in the high-risk group, respectively. Acute genitourinary (GU) toxicities were more common in the MHF than the EHF group, but late gastrointestinal (GI) and GU toxicities did not differ between groups.

Habl et al. reported on the acute toxicity and quality of life from the Ion Prostate Irradiation (IPI) trial in 2016 (ClinicalTrials.gov, Bethesda, MD, USA, no. NCT01641185) [12]. This was a Phase II trial which randomized patients between PBT and carbon ion therapy, both being hypofractionated [13]. A total of 92 patients were enrolled, 46 of which were treated with PBT prescribed with 66 Gy (RBE) in 20 fractions. The PBT patients were treated with pencil beam scanning (PBS) techniques and consisted of low-risk (17%), intermediate-risk (65%), and high-risk (17%). This study had a median follow-up of 22.3 months, and because of this short time the investigators were unable to report on prostate-specific antigen (PSA) progression-free survival, overall survival or late toxicities. Acute GI toxicities of grade 3 were seen in two patients (4.3%), and no patients developed acute GU toxicities of grade 3/4.

Also in 2016, Vargas et al. reported on a randomized clinical trial comparing hypofractionated to standard fractionation PBT for low-risk patients (ClinicalTrials.gov, Bethesda, MD, USA, no. NCT01230866) [14]. Longer follow-up data was later published in 2018 [15,16]. Forty-nine patients received 38 Gy (RBE) in five fractions. The rate of late GU and GI grade 2 or higher toxicities at four-years were 30.4% and 19.6%, respectively, and no grade 3 or higher toxicity was seen. The authors did not present any recurrence results. The incidence of adverse events was observed to be similar between the standard and hypofractionated patients.

In 2017 Henderson et al. reported on the five-year outcomes from a prospective trial accruing 215 prostate patients treated with passively scattered (PS) protons, consisting of low-risk (56%) and intermediate-risk (44%) (ClinicalTrials.gov, Bethesda, MD, USA, no. NCT00693238) [17]. Low-risk patients were prescribed 70 Gy (RBE) in 28 fractions while intermediate-risk patients were prescribed 72.5 Gy (RBE) in 28 fractions. However, a reduction in prescription dose to a minimum of 67.5 Gy (RBE) was performed, if needed, to meet organ at risk constraints. The five-year biochemical control rates were 98.3% in the low-risk patients and 92.7% in the intermediate-risk patients. The rate of late GU and GI grade 3 or higher toxicities at five-years were 1.7% and 0.5%, respectively.

Khmelevsky et al. performed a randomized trial investigating the use of hypofractionated PBT to boost the prostate compared to a photon boost in 2018 [18]. One hundred and fourteen patients consisting of high- and intermediate-risk of progression were treated with the PBS proton boosts. The PBT boost fractionations were either eight fractions of 3 Gy (RBE) per fraction; five fractions of 4.0 Gy (RBE) per fraction; or three fractions of 5.5 Gy (RBE) per fraction, after a photon irradiation of 44 Gy in 22 fractions to the small pelvis. The five- and 10-year biochemical control rates were 60.0% and 55.9%, respectively. Late GI grade 3–4 toxicities were seen in 0.9% of patients, and late GU grade 3–4 toxicities were seen in 2.8% of patients. The authors concluded that the PBT boost significantly reduced early and late rectitis severity when compared to the photon control cases, and no significant difference was observed between the different fractionations.

In 2018 Nakajima et al. compared hypofractionated to standard fractionation PBT [19]. Two hundred and seventy-two patients were treated with hypofractionation, low-risk patients were prescribed 60 Gy (RBE) in 20 fractions, and intermediate- and high-risk patients were prescribed 63 Gy (RBE) in 21 fractions. Most patients received PS PBT, however 6.1% of patients received PBS. Late toxicities and outcomes were not reported due to the patient follow-up of six months. The incidence of acute grade 2 GU toxicities was 5.9%, and no grade 2 GI toxicities were observed in the hypofractionated group. Compared to the standard fractionation, the incidence of GU toxicities was reduced for the hypofractionated patients, and no difference was observed for the incidence of GI toxicities.

Grewal et al. in 2019 reported on the four-year outcomes of a clinical trial accruing 184 patients [20]. The patients consisted of low-risk (10%), favorable intermediate-risk (42%), and unfavorable intermediate-risk (48%). All patients were prescribed 70 Gy (RBE) in 28 fractions and treated with either PS PBT or PBS. The 4-year biochemical control rates were 93.5%. The four-year incidence rate of late GI grade 2 toxicities were 13.6%, and late GU grade 2 toxicities were 7.6%. There was one case of late grade 3 GI toxicity in a patient who had radiation proctitis.

In 2019 Kubes et al. reported on the use of extreme hypofractionation of 36.25 Gy (RBE) in five fractions for early-stage prostate cancer [21]. 200 patients were treated with PBS, 46.5% of which were low-risk and 53.5% were intermediate-risk. The incidence of late GI toxicity of grade 2 was 5.5%, and late GU toxicity of grade 2 was 4%. No grade 3 or higher toxicities were observed. PSA relapse was observed in one patient (1.08%) in the low-risk group and in seven patients (6.5%) in the intermediate-risk group.

Also in 2019, Slater et al. presented the results of a cohort of 145 low-risk patients who were treated with 60 Gy (RBE) in 20 fractions (ClinicalTrials.gov, Bethesda, MD, USA, no. NCT00831623) [22]. The three- and five-year biochemical control rates were 99.3% and 97.9%, respectively. The three-year actuarial rate of grade 2 GU toxicities was 9.5% and grade 2 GI toxicities was 5.1%. One patient had a late grade 3 GU toxicity and there were no grade 3 GI toxicities.

More recently, Vapiwala et al. in 2021 presented a multi-institutional analysis of low- and intermediate-risk patients with fractionations between 2.5 to 3 Gy per fraction and that were treated with either IMRT or PBT [23]. A total of 568 patients received PBT from four institutions with either uniform scanning (US) or PBS. The rate of late GU toxicities of grade 2 and grade 3 was 15.0% and 1.6%, respectively. The rate of late GI toxicities of grade 2 and grade 3 was 11.1% and 0.4%, respectively. The authors concluded that there was no statistical significance in late toxicities between the treatment modalities after adjustment for patient and treatment factors.

**Table 1 cancers-14-02271-t001:** Prostate cancer.

Reference	D	D/#	Stage	n	Age	F	Local Control	Late Toxicity ≥ Grade 2
Johansson 2012 [8], Johansson 2019 [9]	22 *	5.5	Low, intermediate, high and very high risk	504	66	9.4	low-risk 100% at five years Intermediate-risk 94% at five years High-risk 82% at five years Very high-risk 72% at five years	G3 GI 0% G3 GU 2%
Kim 2013 [10], Ha 2019 [11]	60	3	Low, intermediate and high risk	82	66	7.5	76% at seven years	G2 GI 15% G3 GI 4% G2 GU 12%
	54	3.6			69			
	47	4.7			71			
	35	7			67		46% at seven years	G2 GI 13% G2 GU 7%
	35	7			70			
Habl 2014 [13], Habl 2016 [12]	66	3.3	Low, intermediate and high risk	46	69	1.9	100% at 22.3 months	--
Vargas 2016 [14], Vargas 2018 [15,16]	38	7.6	Low-risk	49	65	3	--	G2 GI 19.6% G2 GU 30.4%
Henderson 2017 [17]	70	2.5	Low risk	215	65	5.3	98.3% at five years	G3 GU 1.7%
	72.5	2.5	Intermediate risk				92.7% at five years	G3 GI 0.5%
Khmelevsky 2018 [18]	24 *	3	High and intermediate risk	114	66.9	5.7	60.0% at five years	G2 GI 10.2% G2 GI 0.9% G2 GU 8.3% G3 GU 2.8%
	20 *	4						
	16.5 *	5.5						
Nakajima 2018 [19]	60–63	3	Low, intermediate and high risk	272	69	0.5	-	-
Grewal 2019 [20]	70	2.5	Low-to intermediate-risk	184	-	4.1	93.5% at four years	G2 GI 13.6% G2 GU 7.6%
Kubes 2019 [21]	36.25	7.25	Low-to intermediate-risk	200	64.3	3	99% at three years for low risk 93.5% at three years for intermediate risk	G2 GI 5.5% G2 GU 4%
Slater 2019 [22]	60	3	Low-risk	146	65	3.5	99.3% at three years 97.9% at five years	G2 GI 5.1% G2 GU 9.5% G3 GU 0.7%
Vapiwala 2021 [23]	60–72.5	2.5–3	Low-or intermediate-risk	568	67	3.7	-	G2 GI 11.1% G3 GI 0.4% G2 GU 15% G3 GU 1.6%

D, dose (Gy (RBE)); D/#, dose per fraction (Gy (RBE)); n, number of patients; Age, median age at diagnosis (years); F, median follow-up (years). GI, gastrointestinal; GU, genitourinary. * Hypofractionated PBT utilized as a boost.

### 3.2. Liver

Primary liver cancers are most commonly hepatocellular carcinoma (HCC) or intrahepatic cholangiocarcinoma (ICC). As normal liver parenchyma is considered highly radiation sensitive, techniques permitting safe radiation delivery have been explored. Table 2 summarizes the reported clinical results.

In 2008 Fukumitsu et al. reported a retrospective analysis of 51 patients treated to 66 Gy (RBE) in 10 fractions [24]. The local control rate at three- and five-years was 94.5% and 87.8%, respectively. All lesions were ≥2 cm away from the porta hepatis (PH) or gastrointestinal tract (GIT). Late toxicity included three patients who developed rib fractures and one patient who developed grade 3 radiation pneumonitis.

In 2011 Bush et al. published a prospective observational study of 76 patients treated with PBT to 63 Gy (RBE) in 15 fractions [25]. Of all lesions included, 48% were greater than 5 cm in diameter. Although local control was not reported, the median progression free survival was 36 months and the average onset of local failure was 18 months. Late toxicity was not reported, however five patients of the first 30 treated did experience grade 2 GI toxicity. Those treated subsequently had field margins reduced when the tumor occurred adjacent to the bowel.

In 2011 Mizumoto et al. reported on a study investigating 3 different dose schedules depending on tumor location; protocol A, 66 Gy (RBE) in 10 fractions for lesions not adjacent to the GIT or PH; protocol B, 72.6 Gy (RBE) in 22 fractions for lesions within 2 cm of the PH; protocol C, 77 Gy (RBE) in 35 fractions for lesions within 2 cm of the GIT [26]. In total, 266 patients treated with PS PBT were included for analysis. For the entire cohort, the local control rate at three- and five-years were 87% and 81%, respectively. Ten patients experienced grade 2 or greater late toxicity; three developed rib fractures, one had grade 3 dermatitis and six had grade 2 or greater GI toxicity.

In 2013 Kanemoto et al. studied 67 patients with HCC retrospectively who were treated with PBT to 66 Gy (RBE) in 10 fractions [27]. This report specifically studied the rate of rib fracture (16.4%), correlating to dose-volume histogram analysis. No disease control data was presented.

In 2016 Hong et al. published a prospective single arm, phase II multi-institutional study [28]. The data for 92 patients with either HCC or ICC were presented. Patients were treated with a risk-adapted approach, with peripheral lesions receiving 67.5 Gy (RBE) in 15 fractions, while central lesions were treated with 58.05 Gy (RBE) in 15 fractions. Patients with HCC and ICC had a two-year local control rate of 94.8% and 94.1%, respectively. One patient each developed grade 3 liver failure and grade 3 nonmalignant ascites.

In 2017 Kim et al. published a retrospective study reporting the outcomes of 71 patients with HCC [29]. Of the cohort, 77.5% had recurrent or persistent disease following prior therapy with non-radiotherapy techniques. All patients received 66 Gy in 10 fractions using PS PBT. The three-year local progression free survival rate was 89.9% and three-year overall survival rate was 74.4%. The authors state that no high-grade acute toxicities and no late GI toxicities were observed.

In 2018 Yeung et al. retrospectively reviewed the data for 37 patients with either HCC or ICC that were treated with PBS receiving a median dose of 60 Gy (RBE) in 15 fractions [30]. This study specifically investigated the rate of chest wall toxicity and therefore did not evaluate disease control data. The grade 2 rate of chest wall pain was 19% whilst 11% experienced grade 3 chest wall pain.

In 2020 Parzen et al. evaluated a multi-institutional prospective registry, presenting the data for 63 patients treated with PS PBT and PBS PBT [31]. Although a range of dose and fractionation schedules were used, the median total dose was 58.05 Gy (RBE). The 2-year local control rate was 81.1% at 2 years. For patients with HCC diagnosis, median overall survival was 16.9 months, while ICC patients had median overall survival of 20.1 months. High grade toxicities were rare, with only 1 patient each reporting either grade 3 sinus bradycardia, abdominal pain, hyperbilirubinemia or back pain.

In 2020 Kim et al. studied 45 patients with HCC prospectively who all received 70 Gy (RBE) in 10 fractions with PS PBT [32]. The local progression free survival rate at three years was 95.2% and the three-year overall survival rate was 86.4%. No patients experienced high grade acute toxicity. Late toxicity data was not presented; however, the authors did note that no patient experienced late GI toxicity.

In 2020 Smart et al. retrospectively identified 66 patients who received treatment for ICC with PS PBT, where the median dose delivered was 58.05 Gy (RBE) in 15 fractions [33]. The disease local control rate at two-years was 84% and the two-year overall survival rate was 58%. Although late toxicity was not presented, the rate of high-grade acute toxicity was 11%. One patient had radiation-induced liver damage requiring glucocorticoid treatment.

**Table 2 cancers-14-02271-t002:** Liver cancer.

Reference	D	D/#	Diagnosis	n	Age	F	Local Control	Late Toxicity ≥ Grade 2
Fukumitsu 2008 [24]	66	6.6	HCC	51	-	2.8	94.5% at three years 87.8% at five years	Rib fracture 5.8% G3 lung 2%
Bush 2011 [25]	63	4.2	HCC	76	63 (mean)	-	Median PFS three years	-
Mizumoto 2011 [26]	66 72.6 77	6.6 3.3 3.5	HCC	266	70	-	87% at three years 81% at five years	Rib fracture 1.1% G2 GI 1.1% G3 GI 1.1%
Kanemoto 2013 [34]	66	6.6	HCC	67	69	2.3	-	Rib fracture 16.4%
Hong 2016 [28]	67.5 58.05	4.5 * 3.87 **	HCC, ICC	92	68	1.6	94.8% at two years (HCC), 94.1% at two years (ICC)	G3 blood 1% G3 GI 1%
Kim 2017 [29]	66	6.6	HCC	71	63	2.6	89.9% LPFS at three years	-
Yeung 2018 [30]	60	4	HCC, ICC	37	66	0.9	-	G2 chest wall 19% G3 chest wall 11%
Parzen 2020 [31]	58.05	3.87	HCC, ICC	63	69	0.4	81.1% at two years	G3 cardiac 2% G3 GI 2% G3 investigations 2% G3 MS 2%
Kim 2020 [32]	70	7	HCC	45	63	2.9	95.2% LPFS at three years	-
Smart 2020 [33]	58.05	3.87	ICC	66	76	1.75	84% at two years	-

D, dose (Gy (RBE)); D/#, dose per fraction (Gy (RBE)); n, number of patients; Age, median age at diagnosis (years); F, median follow-up (years). HCC, hepatocellular carcinoma; ICC, intrahepatic cholangiocarcinoma; PFS, progression free survival; LPFS, local progression free survival. * peripheral lesion, ** central lesion.

### 3.3. Lung

In 2006, Nihei et al. reported on 37 patients with medically inoperable (or those that refused surgery) stage 1 non-small cell lung cancer (NSCLC), treated with PBT to 70–94 Gy (RBE) in 20 fractions [35]. With a median follow-up period of two years, the two-year local progression-free and OS rates were 80% and 84%, respectively. The two-year locoregional relapse-free survival rates in Stage IA and Stage IB were 79% and 60%, respectively. Higher rates of grade 3 lung toxicity were observed in patients with larger tumors (Stage IB).

Hata et al. reported an early series of 21 patients with stage 1 NSCLC treated with hypofractionated high dose PBT at the University of Tsukuba [36]. Patients had a median age of 74 years and had a median follow-up of 25 months with local progression-free and disease-free rates of 95% and 79% at two years, respectively. The two-year overall and cause-specific survival rates were 74% and 86%, respectively. No therapy-related toxicity of grade > 3 was observed.

In 2010, Nakayama and colleagues reported on the outcomes of 55 patients treated with PS PBT, median age 77, with T1 and T2 NSCLC treated to one of two dose levels depending on whether they were centrally or peripherally located [37]. OS and progression-free rates were 97.8% (93.6–102.0%) and 88.7% (77.9–99.5%), respectively. Local control of all tumors at two years was 97.0% (91.1–102.8%). Only two patients (3.6%) had grade 3 pneumonitis.

In 2010, Iwata et al. evaluated the clinical outcomes of 80 patients (median age, 76 years) with Stage 1 NSCLC treated with PBT or carbon-ion therapy (57 with PBT and 23 with carbon-ion therapy) using three treatment protocols (two proton and one carbon) [38]. The proton arm delivered 80 Gy (RBE) in 20 fractions or 60 Gy (RBE) in 10 fractions. After a median follow-up of 35.5 months, three-year overall survival, cause-specific survival, and local control rates were 75% (IA: 74%; IB: 76%), 86% (IA: 84%; IB: 88%), and 82% (IA: 87%; IB: 77%), respectively, for all 80 patients combined. The authors noted no significant differences in treatment results among the three protocols and only one patient experienced grade 3 pulmonary toxicity in the 80 Gy proton arm.

In 2011, Chang et al. from the University of Texas M.D. Anderson Cancer Center reported on a series of 18 patients with medically inoperable T1N0M0 (central location) or T2-3N0M0 (any location) NSCLC that were treated with PBT (passive scattering) at 87.5 Gy (RBE) at 2.5 Gy/fraction in this phase I/II study [39]. Four patients had T1 lesions, 13 patients (73%) had T2 lesions, and 15 (83%) had central lesions. At a median follow-up time of 16.3 months, rates of local control were 88.9%, regional lymph node failure was 11.1%, and distant metastasis was 27.8%.

In 2012, Westover and colleagues from Massachusetts General Hospital reported on 15 patients with 20 early stage NSCLC tumors treated with PS PBT to 42 to 50 Gy (RBE) in three to five fractions between 2008 and 2010 [40]. With a median follow-up of 24.1 months, the two-year overall survival and local control rates were 64% and 100%, respectively.

Bush and colleagues from Loma Linda University Medical Center updated their previous reports on a series of patients with stage 1 medically inoperable lung cancer who received hypofractionated PBT with a sequential dose escalation of 51, 60 and 70 Gy (RBE) [41]. Improved OS was demonstrated with increasing dose level (51, 60, and 70 Gy (RBE)) with a 4-year OS of 18%, 32%, and 51%, respectively. At four years, for peripherally located T1 tumors, local control was 96%, disease-specific survival was 88%, and OS was 60%. For larger T2 tumors there was a trend towards improved local control and survival with the 70 Gy (RBE) compared to lower doses. Given these findings, 70 Gy (RBE) is standard for T1 tumors at their institution. Interestingly, no patients experienced pneumonitis, however four patients with peripherally located tumors did experience treated induced chest wall pain.

In 2014, Kanemoto and colleagues from the University of Tsukuba published their experience on 74 patients (median age, 75) treated with high dose PBT for 80 centrally and peripherally located stage 1 NSCLC between 1997 and 2011 [27]. Two protocols were used: 72.6 Gy (RBE) in 22 fractions for centrally located tumors, and 66 Gy (RBE) in 10 or 12 fractions for peripherally located tumors. The median follow-up was 31 months. The 3-year local control rate was reported at 81.8%, however for 1B tumors it decreased (67% for Stage 1B versus 86.2% for Stage 1A). There was a difference in three-year local control rates for centrally located tumors (63.9%) compared to peripherally located tumors (88.4%). Overall survival, disease-specific survival, and progression-free survival rates were 76.7%, 83.0%, and 58.6% at three years, respectively.

Korean investigators Lee et al. published on 55 patients treated with PS PBT, with a variety of fractionation schedules of 50–72 Gy (RBE) in 5–12 fractions for stage 1 (n = 42) and recurrent NSCLC (n = 13) [42]. After a median follow-up of 29 months, 24 (43.6%) patients died, 11 of which were from disease progression. Local control at three years was 94% for T1, and 65% for T2 tumors. Seven patients had local progression with a median time to local progression of 9.3 months. One grade 5 treatment-related adverse event occurred in a patient with symptomatic idiopathic pulmonary fibrosis.

In 2017, Ono et al. from Southern Tohoku PBT Center reported on a series of 20 patients (median age, 75) with central lung cancers located less than 2 cm from the trachea, mainstem bronchus, or lobe bronchus [43]. Various NSCLC stages were included in this treated cohort: stage 1 (75%), stage 2 (20%), and stage 3 (5%). All patients received 80 Gy (RBE) in 25 fractions with PS PBT over five weeks between January 2009 and February 2015. With a median follow-up of 7.5 months, the two-year OS and local control rates were 73.8% and 78.5%, respectively. No grade 3 or higher toxicities were observed.

In 2019, Nakamura and colleagues retrospectively reviewed 39 patients who received hypofractionated PS PBT for centrally located (defined a tumor within 2 cm of the proximal bronchial tree) cT1-2N0M0 lung cancer between 1999 and 2015 [44]. Twenty-four patients (62%) were treated with 80 Gy (RBE) in 20 fractions, whereas eight (21%) were treated with 66 Gy (RBE) in 10 fractions. At a median follow-up period of 48 months, the two-year progression-free survival and overall survival rates were 86% and 100% for T1 disease and 56% and 94% for T2 disease, respectively. Local failure was observed in six patients (27%), regional in seven (32%), distant in seven (32%), and local and distant in two (9%).

In 2019, Badiyan and colleagues from 8 institutions reported on clinical outcomes and toxicities on the largest series to date of patients with recurrent lung cancer (n = 79), the majority of which previously received conventionally fractionated radiation therapy, who were then re-irradiated with PBT [45]. PS PBT, US and PBS techniques were utilized. Of the cohort, 31 patients (39%) received hypofractionated PBT with 2.1–7 Gy/fraction to a median equivalent dose in 2 Gy fractions (EQD2) of 60.4 Gy. After a median follow-up of 10.7 months after PBT, the median OS and progression-free survival was 15.2 months and 10.5 months, respectively. Acute and late grade 3 toxicities occurred in 6% and 1%, respectively. Three patients died after PBT from possible radiation toxicity.

In 2020, Kharod and colleagues reported on 23 patients with T1-T2N0M0 NSCLC treated between 2009 and 2018 using image-guided hypofractionated double-scattering PBT to 60 Gy (RBE) in 10 fractions [46]. After a median follow-up of 3.2 years, OS rates at three and five years were 81% and 50% (95% CI, 27–79%), respectively. Cause-specific survival rates at three and five years were 81% and 71% (95% CI, 46–92%), and the three-year local, regional, and distant control rates were 90%, 81%, and 87%, respectively. Two patients (9%) experienced late grade 3 toxicities.

On behalf of the Proton Collaborative Group, Hoppe et al. reported safety data in 2020 from the first multicenter phase 1 trial investigating hypofractionated PBT in patients with stage II and III non-small cell lung cancer delivered with concurrent chemotherapy [47]. PS PBT, US and PBS techniques were utilized. A stepwise 5-2 dose-intensification protocol design allowed for the four treatment arms: (1) five patients at 2.5 Gy (RBE) per fraction to 60 Gy (RBE); (2) five patients at 3.0 Gy (RBE) per fraction to 60 Gy (RBE); (3) seven patients at 3.53 Gy (RBE) per fraction to 60.01 Gy (RBE); (4) One patient at 4.0 Gy (RBE) per fraction to 60 Gy (RBE) (1 patient). Ultimately, 18 patients were recruited prior to the trial closing due to poor accrual, with no maximum tolerated dose identified. Two patients treated at 3.52 Gy (RBE) experienced a serious adverse event due to chemotherapy. No dose-limited PBT associated serious adverse events were reported.

Table 3 summarizes the reported clinical results.

**Table 3 cancers-14-02271-t003:** Lung cancer.

Reference	D	D/#	Stage	n	F	Age	Local Control	Late Toxicity ≥ Grade 2
Nihei 2006 [35]	70–94	3.5–4.9	46% T1 54% T2	37	2	75	80% at two years	G3 lung 8% G2 lung 8%
Hata 2007 [36]	60	6	T1N0M0, Stage IA, 11. T2N0M0, Stage IB, 10.	21	2.1	74	2 yr T1 100% 2 yr T2 90% 95% overall	G3 0% G2-2 patients (subcutaneous induration, myositis)
Nakayama 2010 [37]	66 * or 72.6 **	3.3–6.6	T1/T2 52%/48%	55	1.5	77	97% at two years	G3 lung 3.6% G2 lung 3.6%
Iwata 2010 [38]	80/20 or 60/10	4 or 6	42 T1 38 T2	80	3	76	three yr T1 87, T2 77	G3 lung 1 patient (in PBT arm to 80 Gy (RBE)) G2 lung 13% G2 skin 16% G2 rib fracture 23% G2 soft tissue 6%
Chang 2011 [39]	87.5	2.5	T1 4 T2 13 T3 1	18	1.4	74	88.9%	G3 skin 17% G2 skin 67% G2 fatigue 44% G2 lung 11% G2 esophagitis 6% G2 chest wall pain 6%
Westover 2012 [40]	42–50	10–16	T1a 16 (80%) T1b 2 (10%) T2a 2 (10%)	15	2	78	100% at two years	G2 skin 5% G2 chest wall pain 5% G2 fatigue 5% G3 lung 5%
Bush 2013 [41]	51–70	5.1–7	T1 47 T2 64	111	4	73	At four years, T1 86–91% T2 45–74%	G3 rib fracture (4 patients) No reported pneumonitis no other treatment-related adverse events of grade 2 or higher reported
Kanemoto 2014 [27]	66 * or 72.6 **	3.3 to 6.6	Stage IA 59 (74%) Stage IB 21 (26%)	74	2.6	75	The three-year local control rate was 86.2% for stage IA tumors and 67.0% for stage IB tumors	G4 bone 13.8% G3 lung 1.3% G3 skin 1.3% G2 skin 2.5% G2 esophagus 1.3%
Lee 2016 [42]	50.0–72.0	6–12	T1a/T1b/T2 19/20/16 (35%/36%/29%)	55	2.4	75	three-year T1 94%, T2 65%	G2 rib fracture 5.4% G2 pneumonitis 12.7%
Ono 2017 [43]	80	3.2	Stage 1 (75%), Stage 2 (20%), Stage 3 (5%) Central lung tumours	20	2.3	75	78.5% at 2 years	G2 lung 10% G2 bone 10%
Badiyan 2019 [45]	40–63	2.1–7	Recurrent lung cancer post prior radiation therapy	31	0.9	69	Median local relapse-free survival was 12.9 months (95% CI, 10.4–15.4); 6-, 12-, and 18-month rates were 77.4%, 56.3%, and 30.9%, respectively for all patients	G3 fatigue (n = 1) G2 dyspnea 5% G2 pneumonitis (n = 1)
Nakamura 2019 [44]	66–80	4–6.6	T1 21 (54%) T2 18 (46%)	39	4	75	73%	G3 lung 3% G2 lung 10% G2 skin 3%
Kharod 2020 [46]	60	6	T1-T2N0M0 NSCLC (T1, 46%; T2, 54%)	23	3.2	74	90% at three years	G3 9% (including 1 patient who developed bronchial stricture) G2 34% non-hematologic
Hoppe 2020 [47]	60	2.5–4.0	Stage II or III NSCLC; concurrent chemotherapy	18	-	71	-	G4 pneumonitis 5.6% (3.53 Gy per fraction arm) G2 pneumonitis 11.1%

D, dose (Gy (RBE)); D/#, dose per fraction (Gy (RBE)); n, number of patients; F, median follow-up (years); Age, median age at diagnosis (years); non-small cell lung cancer (NSCLC). * peripheral tumor, ** central tumor.

### 3.4. Other Sites

Studies investigating the clinical effects of hypofractionation in all sites outside those discussed above are presented below. The majority of studies in this section relate to intracranial sites and breast cancer. Table 4 and Table 5 summarize the reported clinical results for intracranial sites and breast cancer, respectively.

#### 3.4.1. Intracranial

In 2001, Vernimmen et al. reported on the experience of treating 23 patients with skull base meningiomas [48]. Eighteen of the 23 patients were treated with a hypofractionated regimen of three fractions, while five patients received 16 or more fractions. The mean ICRU reference dose in the hypofractionated regimen was 20.3 Gy (RBE) while the ICRU reference doses in the more fractionated regimen ranged from 54 Gy (RBE) in 27 fractions to 61.6 Gy (RBE) in 16 fractions. The mean target volume for the hypofractionated regimen was 15.6 cm^3^. In the hypofractionated cohort, 16 of 18 (89%) patients remained clinically stable or improved while two of 18 (11%) deteriorated. Radiologic control was achieved in 88% of patients, while two patients had a marginal failure. The group concluded that hypofractionation of PBT was effective and safe in controlling large and complex-shaped skull-based meningiomas. Using information from the different fractionation regimens, the authors estimated an α/β value of 3.7 Gy for meningiomas, suggesting a scenario of low tumor α/β.

The same group also reported on the long-term results of stereotactic proton beam radiotherapy for acoustic neuromas [49]. In this study, 51 patients were treated with hypofractionation in a 26 Gy (RBE) in a three fraction regimen and followed for a minimum of two years. The five-year results showed a 98% local control, with hearing preservation of 42%, facial nerve preservation of 90.5% and trigeminal nerve preservation of 93%. The group concluded that hypofractionated PBT offers long-term control with minimal side-effects for patients with large, inoperable acoustic neuromas.

Ryttlefors et al. reported on long-term tumor control following hypofractionated PBT for intracranial benign meningioma [50]. The treatment regimen consisted of 6 Gy (RBE) fractions to a total dose of 24 Gy (RBE). Seventeen of 19 (89%) patients experienced tumor growth arrest which was deemed to be comparable (vs. 98%) with a large series treated with conventional fractionation, as reported by Slater et al. [51]. One patient in the cohort experienced edema in the adjacent brainstem suggestive of a late treatment- induced toxicity. The authors suggest that the risk of late complications appears to be similar with conventional fractionation, but this needs to be validated in a larger cohort.

In 2014 Kim et al. reported on an investigation to determine an optimal fractionation schedule for PS PBT of chordoma [52]. While the total dose was varied across 20 patients, dose per fraction was kept constant at 2.4 Gy (RBE). Superior five-year control was observed in patients prescribed greater than 70 Gy (RBE) total dose and no serious skin toxicity was reported in patients who received more than 1 beam in the treatment plan. The authors did not report any other toxicity events.

Vlachogiannis et al. reported on their experience treating 170 patients with WHO grade I meningiomas in a hypofractionated regimen of 5 Gy (RBE) or 6 Gy (RBE) per fraction with PS PBT [53]. Actuarial five- and 10-year progression free survival rates were 93% and 85%, respectively. Radiation related complications were seen in 16 patients (9.4%), with pituitary insufficiency being the most common. The authors noted that tumor location in the anterior cranial fossa was the only factor to significantly increase the risk of complication and suggest that PBT is particularly well suited to larger WHO grade I meningiomas.

Cao et al. presented a dosimetric study comparing different hypofractionated stereotactic radiotherapy techniques in treating intracranial tumors > 3 cm in longest diameter [54]. Treatment plans were generated with Gamma Knife, Cyberknife, non-conformal arc VMAT, conformal arc VMAT, passive scattered PBT and intensity modulated PBT. The authors suggest that PBT may represent a desirable alternative to advanced photon techniques for large and irregularly shaped target volumes close to critical structures, but that this should be assessed on a case-by-case basis.

**Table 4 cancers-14-02271-t004:** Intracranial.

Reference	Cancer Type	D	D/#	Stage	n	F	Age	Local Control	Late Toxicity
Vernimmen 2001 [48]	Skull base meningioma	20.3	6.77	-	18	40 months (clinical) 31 months (radiological)	55	88% at five years	11% (n = 2) transient cranial nerve neuropathy
Vernimmen 2009 [49]	Acoustic Neuroma	26	8.67	-	51	10 years	50	96% at five years	8.3% (n = 4) VIIth nerve neuropathy 8.3% (n = 4) Vth nerve neuropathy low grade
Kim 2014 [52]	Chordoma	64.8–79.2	2.4	-	20	43 months	53	-	n = 1 grade 3 rectal bleeding for sacral chordoma patient
Ryttlefors 2016 [50]	Skull base meningioma	24	6	WHO grade I	19	11.6 years (MR)	52	89% at median 11.6 years	5.3% (n = 1) brainstem oedema
Vlachogiannis 2017 [53]	Meningioma	14–46	5,6	WHO grade I	170	84 months	54	93% at five years and 85% at 10 years progression free survival	7.4% (n = 6) pituitary insufficiency in patients with significant dose to pituitary 2.9% (n = 5) signs of radiation necrosis 4.4% (n = 5) visual impairment in patients with significant dose to optical structures

D, dose (Gy (RBE)); D/#, dose per fraction (Gy (RBE)); n, number of patients; F, median follow-up; Age, median or mean (years).

#### 3.4.2. Breast

Kim et al. reported on a phase 2 trial of accelerated, hypofractionated whole-breast irradiation delivered as a daily dose of 3 Gy (RBE) to the whole breast followed by a tumor bed boost of 9 Gy (RBE) delivered in three fractions [55]. The study included 276 patients (pT1-2 and pN0-1a). With a median follow-up of 57 months, the rate of five-year locoregional recurrence was 1.4% and disease-free survival was 97.4%. No grade 3 skin toxicity was reported during the follow-up period.

Smith et al. reported on reconstruction outcomes and predictors of complications with post-mastectomy PBS PBT after immediate breast reconstruction [56]. In a study of 51 women, 37 (73%) received conventional fractionation (median 50 Gy in 25 fractions) while 14 (27%) received hypofractionation (median 40.5 Gy in 15 fractions). Among irradiated breasts, hypofractionation was significantly associated with reconstruction failure and the authors concluded that optimal dose-fractionation after immediate breast reconstruction is needed.

Mutter et al. reported an initial experience in the use of a heterogeneous RBE-based biological dose constraint for the brachial plexus in the context of a randomized trial of conventional versus hypofractionated postmastectomy PBS PBT [57]. The biological dose assumed a linear relationship between RBE and dose-averaged LET. Following the introduction of the biological dose constraint for the brachial plexus, mean physical dose to the brachial plexus was reduced. The study reported three-month arm symptoms for patients planned with and without the use of biological constraints to the brachial plexus. The authors acknowledge that further follow-up is needed for conclusive findings.

**Table 5 cancers-14-02271-t005:** Breast cancer.

Reference	Cancer Type	D	D/#	Stage	n	F	Age	Local Control	Late Toxicity
Kim 2013 [55]	Breast	Whole breast = 39 Tumor bed boost = 9	3	Early stage	276	57 months	53	97.4% disease free survival	No patient with Grade II or greater toxicity at 2 years or after
Smith 2019 [56]	Breast	40.5	15	Stage I–Stage III. Post mastectomy	51 (conventional and hypofractionation)	16 months	-	-	Hypofractionation significantly associated with reconstruction failure
Mutter 2020 [57]	Breast	40	2.67	Stage II–III	51 (conventional and hypofractionation)	24 months	38–65 (range)	-	-

D, dose (Gy (RBE)); D/#, dose per fraction (Gy (RBE)); n, number of patients; F, median follow-up; Age, mean or median (years).

#### 3.4.3. Miscellaneous

A number single case studies of hypofractionated PBT have been reported since 2017 [58,59,60]. Seidensaal et al. published a study protocol to compare the outcomes of hypofractionated PBT vs carbon ion therapy for neoadjuvant irradiation of retroperitoneal soft tissue sarcoma [61]. Nam et al. reported on their institutional experience in moderately hypofractionated re-irradiation of nasopharyngeal cancer, with a small percentage of patients having PBT as part of their treatment [62]. Yang et al. evaluated toxicities of moderately hypofractionated proton craniospinal irradiation for leptomeningeal metastases [63]. The authors concluded that this was a safe treatment option for these patients.

## 4. Conclusions

Hypofractionation has been increasingly utilized in X-ray EBRT. With the improved healthy tissue sparing capable with PBT it is possible that hypofractionation will be better tolerated with PBT. In the last five years there have been 36 reported clinical follow-up data involving hypofractionation with PBT, making over 60% of the total currently reported data. This shows a significant and growing interest in the field of hypofractionation with PBT, where the most common reported anatomical sites were lung, prostate and liver. The results of the current reported data show that in many cases hypofractionated PBT can be well tolerated.

In the case of prostate and liver cancer patients, doses per fraction of up to 7.6 Gy (RBE) have been investigated. The reported outcomes were that the hypofractionation was well tolerated with a similar incidence of adverse events between standard and hypofractionated patients. In the case of lung cancer patients, doses per fraction of up to 16 Gy (RBE) have been reported. Similarly, studies have reported that hypofractionation was well tolerated. Even at the high dose per fraction of 16 Gy (RBE), a low pulmonary toxicity was observed which was attributed to the reduced integral dose to surrounding lung from PBT.

With the use of hypofractionation, PBT may be made more cost effective and may improve the accessibility of PBT to more patients. Future clinical trials are still needed to determine the optimal fractionation regime for many anatomical sites. The majority of reported outcomes for all treatment sites were involving PS, however with increasing utilization of PBS it will be important to investigate whether this results in improved patient outcomes. Additionally, the RBE of PBT is an area which requires further investigation to reduce RBE uncertainties.

## Figures and Tables

**Figure 1 cancers-14-02271-f001:**
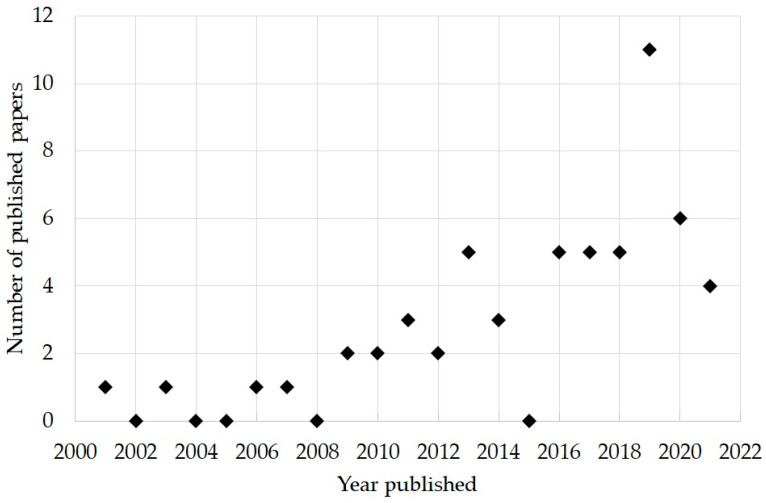
Percentage of publications reporting on clinical outcomes with hypofractionated proton therapy between 2001 and 2021.

**Figure 2 cancers-14-02271-f002:**
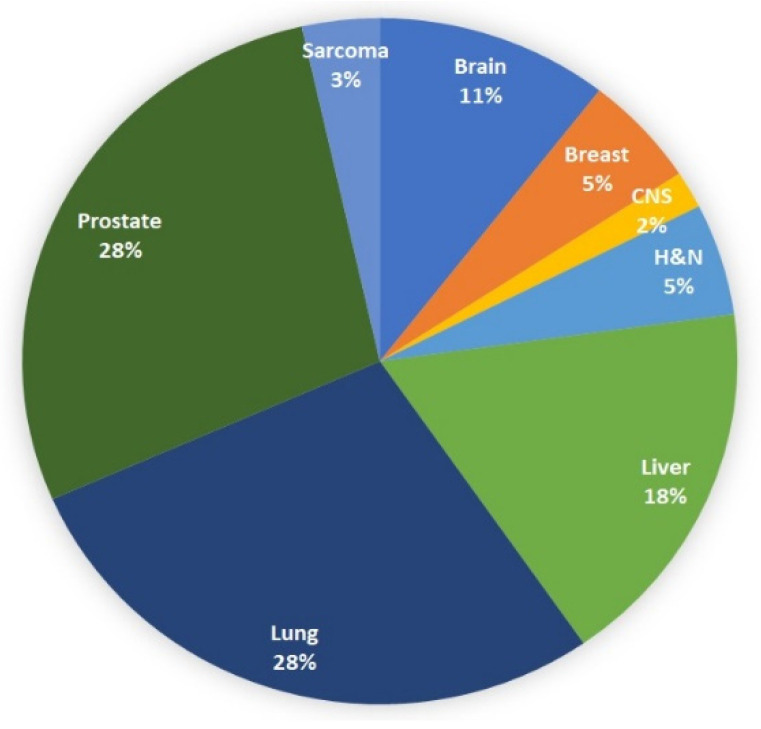
Distribution of treatment sites in reviewed publications.

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
