# Peer review of "The Role of Hypofractionation in Proton Therapy"

_cancers, 2022, doi:10.3390/cancers14092271_

Round 1

Reviewer 1 Report

This review article seems to have included many previously published articles with short summary on each reference.

It is advised for the authors to add summaries on different organ sites (prostate, lung, liver, others) in order to make this review more easily understandable by the potential readers.

In practice, most of partile beam therapy institutes may suffer from longer waiting for the PBT to start than photon treatment, mainly because of limited resource and operation capacity. This is true at the reviewer's institute. It is advised for the authors to add their opinion on how to maintai the efficiency in resource utilization, where hypofractionation could be a reasonable tool to achieve this goal.

Reviewer 2 Report

The authors comprehensively reviewed the clinical data of hypofractionated PBT. While this manuscript is well-organized, some points are needed to be modified before acceptance for publication.

1. Figure 1: Y-axis might be changed from % of published papers to number of published papers.

2. Figure 2: Percentage of each treatment region should be added. In addition, abbreviations (CNS, H&N) should be described.

3. Table 1: GI and GU were not described in the bottom of Table.

4. Table 2: F/U period should be re-checked. More than 30 years of F/U is impossible. Abbreviations (PFS, LPFS) were not described in the bottom of Table.

5. Table 3: Some values are missing, but it could be found in the relevant reference. The authors should carefully re-check the values in the Table.

1) Reference [35]: Age was missing, but I can find “median 75 years (range 63-87)” in the reference. While the authors described G3 lung toxicity 8%, the title of the column is “Late toxicity ≥ grade 2”. Then, the column should be 16.2%. Unless the authors should describe G2 and G3 separately, as reference [44].

2) Reference [37]: Age was also missing, but it can be found in the reference (74 years). While median F/U is 8.1 (years) in the table, it is 17.7 months in the reference. As 2 patients had grade 3 pneumonitis and 2 patients developed grade 2 pneumonitis, “Late toxicity ≥ grade 2” should be 7.3%. Unless the authors should describe G2 and G3 separately, as reference [44].

3) Reference [47]: I can find “Age” in the reference (71 years). While the authors describe only G4 pneumonitis, G2 should be described because the title of the column is “Late toxicity ≥ grade 2”.

6. Please re-check the bibliography carefully. For example, mistyping “–” is found at line 575, 601, 605, 609, 616.

Reviewer 3 Report

It is well written meta-analysis study showing the advantages and disadvantages using proton therapy accelerated shame, but there are no original conclusions.

I strongly recommend mention the used irradiation technique and irradiation mode to every cited paper.

Reviewer 4 Report

This review article summarized published studies on hypofractionated proton beam therapy for prostate cancer, hepatocellular carcinoma, lung cancer, breast cancer, and other cancers. This article is informative and well-structured. This reviewer's concerns are follows.

1. Description of differences between hypofractionated proton beam therapy and hypofractionated photon radiotherapy is lacking, although clinical evidences are not enough. I recommend to describe other potential advantages of hypofractionated proton beam therapy compared to hypofractionated photon radiotherapy besides cost-effectiveness in introduction or conclusions.

2. Radiobiology section is too brief to understand. Please add more description in detail. 

Round 2

Reviewer 2 Report

Generally, the manuscript has been revised properly according to the reviewers’ comments.